



# Exposure dating of detrital magnetite using ³He enabled by microCT and calibration of the cosmogenic ³He production rate in magnetite

Florian Hofmann[1,3], Emily H. G. Cooperdock[2], A. Joshua West[2], Dominic Hildebrandt[3], Kathrin Strößner[3], Kenneth A. Farley[1]

[1]California Institute of Technology, Division of Geological and Planetary Sciences, MC 100-23, 1200 E California Blvd, Pasadena, CA 91125, USA
[2]University of Southern California, Department of Earth Sciences, 3651 Trousdale Parkway, Los Angeles, CA 90089, USA
[3]Ludwig-Maximilians-Universität München, Department of Earth and Environmental Sciences, Luisenstr. 37, 80333 Munich, Germany

*Correspondence to*: Florian Hofmann (fhofmann@caltech.edu)

**Abstract.** We test whether X-ray micro computed tomography (microCT) imaging can be used as a tool for screening magnetite grains to improve the accuracy and precision of cosmogenic ³He exposure dating. We extracted magnetite from a soil developed on a fanglomerate at Whitewater, California, which was offset by the Banning Strand of the San Andreas Fault. This study shows that microCT screening can distinguish between inclusion-free magnetite and magnetite with fluid or common solid inclusions. Such inclusions can produce bulk ³He concentrations that are significantly in excess of expected cosmogenic production. We present Li concentrations, major and trace element analysis, and magnetite (U-Th)/He cooling ages of samples in order to model the contribution from radiogenic, nucleogenic, and cosmogenic thermal neutron production of ³He. We show that mineral inclusions in magnetite can produce ³He concentrations of up to four times that of the cosmogenic ³He component, leading to erroneous exposure ages. Therefore, grains with inclusions must be avoided in order to facilitate accurate and precise magnetite ³He exposure dating. Around 30 % of all grains were found to be without inclusions, as detectable by microCT, with the largest proportion of suitable grains in the grain size range of 400-800 µm. While grains with inclusions have ³He concentrations far in excess of the values expected from existing ¹⁰Be and ²⁶Al data in quartz at the Whitewater site, magnetite grains without inclusions have concentrations close to the predicted depth profile. We measured ³He concentrations in aliquots without inclusions and corrected them for Li-produced components. By comparing these data to the known exposure age of 53.5 ka, we calibrate a magnetite ³He SLHL production rate of 116±13 at g⁻¹ a⁻¹. We suggest that the microCT screening approach can be used to improve the quality of cosmogenic ³He measurements of magnetite and other opaque mineral phases for exposure age and detrital studies.

## 1 Introduction

Since cosmogenic nuclide studies are limited by the minerals present in the available lithology, they benefit from a wide array of potential target phases. The most widely used phase for cosmogenic dating is quartz, using the well-established ¹⁰Be and ²⁶Al systems (e.g. Granger et al., 2013a). Sample preparation, quartz purification, dissolution, and chemical separation of



Be and Al is a laborious, time-consuming, and dangerous task. Measurement by Accelerator Mass Spectrometry (AMS) is expensive compared to conventional mass spectrometry. Radioactive cosmogenic nuclides, such as $^{10}$Be, $^{26}$Al, and $^{36}$Cl, require tens or hundreds of grams of a pure target phase, which has to be extracted from typically several kilograms of bulk

material. In contrast, the relatively higher production rate of $^3$He and the low detection limit using conventional noble gas mass spectrometry allow for measurements on milligram quantities of sample and without the use of harsh chemicals.

We investigate the use of magnetite ($Fe_3O_4$) for $^3$He exposure dating in order to widen the array of potential target phases for cosmogenic studies. Magnetite is very retentive to helium, with a nominal closure temperature around 250 °C (using 10 °C Ma$^{-1}$ and 500 µm grain size; Blackburn et al., 2007), which prevents thermal resetting at surface and near-surface conditions.

It can be easily isolated by magnetic separation with a hand magnet, either directly in the field or from crushed bulk rock or soil samples in the lab. Magnetite forms in a wide range of geologic settings and petrologic conditions, and is a common phase in mafic and felsic igneous rocks (Buddington and Lindsley, 1964), as well as an accessory mineral in many types of metamorphic rocks (e.g. Nadoll et al., 2012). Magnetite is also produced by hydrothermal alteration, which represents an important ore forming process (e.g. Nadoll et al., 2014). Magnetite is therefore likely to be present in many lithologies and

depositional environments, making it a promising target phase for in-situ cosmogenic exposure dating.

Magnetite is resistant to weathering and can also be used for detrital studies. After a recent calibration of the $^{36}$Cl production rate (Moore and Granger, 2019a), detrital magnetite was successfully used as a target phase for deriving watershed-averaged denudation rates (Moore and Granger, 2019b). Magnetite has also been developed as a target phase for cosmogenic $^{10}$Be dating (Granger et al., 2013b; Rogers et al., 2013; Moore, 2017).

The use of magnetite for cosmogenic $^3$He exposure dating was first proposed by Bryce and Farley (2002). Kober et al. (2005) measured $^3$He concentrations in Fe-Ti-oxide minerals (mainly magnetite and ilmenite) from an ignimbrite in Chile and found a good agreement with other cosmogenic systems. They also found that the chemical composition, mainly Ti substitution for Fe and associated phase changes, had no effect on the cosmogenic $^3$He production rate. Matsumura et al. (2014) measured concordant $^{10}$Be, $^{36}$Cl, and $^3$He exposure ages in magnetite, but reported high $^{26}$Al and $^{21}$Ne concentrations

due to silicate inclusions. This observation highlights the contribution of frequently present inclusions within the magnetite, which can potentially increase the measured concentration of cosmogenic nuclides. We will show that solid inclusions in magnetite can have a significant effect on the bulk $^3$He concentrations, which can lead to an overestimation of the exposure age. Mineral inclusions must therefore be avoided in order to enable a robust application of this technique.

Since magnetite is an opaque phase, light microscopy cannot be used to detect inclusions, as is commonly done for

transparent phases, such as olivine (e.g. Trull et al., 1991). We propose that the solution to the problem of inherited $^3$He from inclusions is to screen magnetite grains using X-ray micro computed tomography (microCT) and only select inclusion-free grains for analysis. This approach was successfully used by Cooperdock and Stockli (2016), Cooperdock et al. (2020), and Schwartz et al. (2020) for selecting magnetite crystals for (U-Th)/He thermochronology to avoid the interference caused by other U- and $^4$He-bearing mineral inclusions.



Here, we present measurements of $^3$He concentrations in magnetite aliquots with and without inclusions from a depth profile of a soil at Whitewater (Southern California). We compare these data to existing $^{10}$Be and $^{26}$Al concentrations measured in quartz taken from the same soil samples. We show that using microCT scanning as a tool to select only grains without inclusions leads to $^3$He data, which conforms well to the expected depth profile. We use Li and bulk element concentrations to correct measured $^3$He concentrations for the radiogenic, nucleogenic, and cosmogenic thermal neutron components of $^3$He

production. These data are also used to calibrate the cosmogenic magnetite $^3$He production rate, enabling the future use of the magnetite $^3$He system for cosmogenic exposure studies.

## 2 Materials and Methods

### 2.1 Production of $^3$He in magnetite

The concentration of $^3$He measured in a sample can be attributed to production by several different processes (e.g. Farley et

al., 2006; Amidon et al., 2008; 2009; Dunai et al., 2007), which have to be taken into account to calculate an exposure age (Eq. 1). The cosmogenic (*cosmo*), muogenic (*muon*), nucleogenic (*nucl*), cosmogenic thermal neutron (*CTN*), and radiogenic (*fission*) components accumulate as functions of mainly the exposure age $t_{exp}$, the cooling age $t_c$, the Li concentration $Li$, and the concentrations of radioactive elements $U$, $Th$, and $Sm$.

$$^3He = {}^3He_{cosmo}(t_{exp}) + {}^3He_{muon}(t_{exp}) + {}^3He_{nucl}(t_c, U, Th, Sm, Li) + {}^3He_{CTN}(t_{exp}, Li) + {}^3He_{fission}(t_c, U) \qquad (1)$$

In terrestrial materials $^3$He is primarily produced due to spallation induced by high-energy cosmic-ray neutrons (Kurz, 1986; Lal, 1987), which is generally termed the "cosmogenic" component. Both $^3$He and $^3$H are produced by this process, the latter of which decays to $^3$He with a half-life of around 12 a (Lal, 1987). Cosmic-ray exposure studies primarily use the temporally constant or varying production rate of this component to calculate an exposure age. Cosmic-ray muons also contribute to the production of $^3$He (Nesterenok and Yakubovich, 2016). However, due to the short exposure (<100 ka) and the shallow

depths (<2 m) of the samples we have studied, the high-energy muon component of spallogenic production is negligible and will not be considered here.

There are several estimates for the cosmogenic production rate scaled to sea-level high latitude (SLHL) of magnetite. Theoretical predictions of Masarik and Reedy (1996) for element-specific production rates of 40 at g$^{-1}$ a$^{-1}$ for Fe and 135 at g$^{-1}$ a$^{-1}$ for O lead to a combined SLHL $^3$He production rate by high-energy cosmogenic neutrons in magnetite (Fe$_3$O$_4$) of 66 at

g$^{-1}$ a$^{-1}$. Calibrations by Bryce and Farley (2002) suggest a cosmogenic production rate of 69-77 at g$^{-1}$ a$^{-1}$. Kober et al. (2005) modeled a $^3$He production rate of 122 at g$^{-1}$ a$^{-1}$ using element-specific rates and measured a production rate in Fe-Ti-oxides of 120±12 at g$^{-1}$ a$^{-1}$ based on comparison with other cosmogenic systems.

The energy spectra of $^3$H and $^3$He produced by high-energy neutron-induced spallation are broad (Nesterenok and Yakubovich, 2016), leading to a wide redistribution of $^3$He. The ejection distance distribution of $^3$H has a mode at 56 µm in





magnetite and 120 µm in a matrix equal to the average composition and density of the soil, with 20 and 43 µm, respectively, for $^3$He. Stopping distances were determined using the Stopping Ranges in Matter (SRIM-2013) model (Ziegler et al., 2010). Production of $^3$He can also occur by thermal neutron capture via the $^6$Li(n,α)$^3$H(β$^-$)$^3$He pathway (Andrews and Kay, 1982). One source of thermal neutrons is decelerated cosmic-ray neutrons, called "cosmogenic thermal neutrons" (CTN). Another source of thermal neutrons are α-particles from α-decays as well as spontaneous fission of U and Th (Halpern, 1971), which

can induce (α,n) reactions with matrix elements, mainly Si, O, Al, Mg, Fe (Andrews and Kay, 1982). This component is termed "radiogenic thermal neutrons" (RTN), which produce nucleogenic $^3$He through reactions with $^6$Li. Moderation by matrix elements (mainly B, Sm, Gd, Li, and Cl) also controls the neutron flux at any depth below the surface (Andrews and Kay, 1982; Farley et al., 2006; Dunai et al., 2007; Amidon and Farley, 2009).

The energy of tritium nuclei produced by thermal neutron capture is 2.738 MeV (Biersack et al., 1986), which equals a

stopping range of 22 µm in magnetite and 47 µm in the matrix. Li concentrations of magnetite have previously been reported as 0.2-4 ppm in an ignimbrite (Kober et al., 2005) and 19-22 ppm in a rhyolite (Amidon et al., 2009), the former being negligible and the latter leading to a significant production of non-cosmogenic $^3$He. A characterization of the major and trace element content as well as the Li concentration is therefore critical in assessing the magnitude of the $^3$He production by thermal neutrons.

Another possible source of $^3$He in minerals is $^3$H produced as a result of ternary fission of U (Vorobiev et al., 1969; Halpern, 1971). This component is generally negligible for U concentrations <10 ppm (Farley et al., 2006), but can be an important component when high-eU inclusions, such as apatite and zircon, are present. The kinetic energy of radiogenic $^3$H of 8.1±0.2 MeV (Vorobiev et al., 1969) leads to an average ejection distance of 121 µm within magnetite and 264 µm in the soil.

Cosmogenic and CTN-produced $^3$He accumulates during exposure to cosmic rays, whereas radiogenic and nucleogenic $^3$He

start accumulating at the time when the material is below the magnetite He closure temperature. Knowledge of the cooling age is therefore necessary to assess the latter components.

Magnetite has been observed to host fluid inclusions (e.g. Guzmics et al., 2011), which could potentially contribute amounts of high-Ra helium. However, plutonic rocks spend a sufficiently long time at high temperatures that helium in fluid inclusions is likely to be lost by diffusion (Farley et al., 2006). In this study, we will show through crushing experiments that

there is no measurable amount of $^3$He due to fluid inclusions in the present samples.

Since the material studied here is detrital, production of $^3$He can have occurred before the deposition of the fanglomerate, either during in-situ exposure before erosion or during transport. This production will be by the cosmogenic and CTN pathways described above. There is no rigid a-priori way of estimating this component. Inherited concentrations in clastic sediments, however, are likely to be constant with depth (e.g. Gärtner et al., 2020). We therefore attribute any remaining

depth-constant component to this mechanism after accounting for $^3$He production by all other pathways.





## 2.2 Study site and sampling

We sampled material on a well-studied offset alluvial fan at Whitewater Hill on San Gorgonio Pass, California, USA (Fig. 1a). The deposits represent the surface of an incised alluvial fan, the apex of which has been laterally and vertically offset by the Banning Strand of the San Andreas Fault from its initial position at the mouth of Whitewater Canyon (Owen et al., 2014;

Kendrick et al., 2015; Gold et al., 2015). With an established total offset (Huerta, 2017), an exposure age of the alluvial fan surface can be used to constrain the offset rate of the Banning Strand of the San Andreas Fault. For the same surface, Owen et al. (2014) previously measured an exposure age of $53.9^{+19.0}_{-13.0}$ ka for a depth profile and ages of 5.6-61.0 ka for surface boulders using [10]Be in quartz.

The Cabezon fanglomerate deposits at Whitewater are poorly sorted and are composed of grain sizes from clay to boulder-

size. The lithic clasts are composed mainly of granite, granodiorite, monzonite, and biotite gneiss (Fosdick and Blisniuk, 2018), with grain sizes between 1 cm and 1 m. A reddish soil has developed in the uppermost 2-3 m of the alluvial fan terrace (Fig. 1b). The A horizon and part of the B horizon of the soil have been eroded, most likely by deflation. The fanglomerate is highly grusified, and even large clasts disaggregate with minimal force.

We sampled material on a vertical profile from the surface to 177 cm depth (Fig. 1) at coordinates 33.9340° N, 116.6347° W,

and an elevation of 520 m. About 1 m of material was removed from the vertical face of the outcrop before sampling to reduce the effect of lateral cosmic-ray exposure. All samples were taken over a depth interval of 2 cm, except for the surface sample, which was taken from 0 to 5 cm depth. A sample of the unweathered fanglomerate (17WW-B1) was taken below the zone of soil formation at around 5 m depth below the surface. These samples integrate material from many clasts of varying sizes and lithologies. In a previous study (Hofmann, 2019), quartz was extracted from these samples to measure [10]Be and

[26]Al in quartz, so these data can be directly compared to the measurements of [3]He in magnetite presented here.

## 2.3 Sample processing and microCT scanning

Bulk soil samples (3-5 kg) were dried at room temperature and spread out over the surface of a table. The strongly magnetic fraction was collected with a neodymium hand magnet held about 5 cm above the surface, which yielded several grams of magnetic material for each sample. This magnetic fraction (see Fig. 2) was sonicated for 10 min, washed several times, dried

at 50 °C for 24 h, and sieved to >250 μm. From this separate, individual magnetite grains without obvious intergrowth with other minerals were picked under a light microscope. The resulting separate was again sonicated in water and dried to remove remaining smaller surface impurities and dust. The clay fraction of one sample was also extracted by sieving bulk material to <2 μm.

The mineral phase of this separate was confirmed as magnetite using Attenuated Total Reflection Fourier-Transform

Infrared (ATR-FTIR) spectroscopy. A subset of aliquots of several mg were powdered using a mortar and pestle. ATR-FTIR spectra were obtained from these powdered samples, which were compared to spectra of synthetic magnetite pigments



acquired under the same conditions (Fig. 3). The location and height of characteristic peaks was used to confirm the phase as magnetite and to detect possible contribution from other phases.

A fraction of the picked magnetic separate was directly analyzed for $^3$He without microCT screening, and grains from the
remaining separate were arrayed on 5 mm x 5 mm sheets of paper coated with double-sided adhesive tape (see Fig. 2). An opposing piece of paper without adhesive was placed on top of the mineral grains, and up to eight layers were stacked (see Cooperdock et al., 2020) into an oriented paper box of 5.2 mm x 5.2 mm x 5.2 mm.

These boxes were scanned at the University of Southern California Molecular Imaging Center using a Scanco µCT50 (120 kV, 100 µA, 3.3 µm resolution). The resulting volumes were analyzed and segmented using 3D Slicer (Fedorov et al., 2012).
Inclusions within magnetite grains were identified via intensity on the reconstructed images. Each magnetite grain was classified as one of four categories: (A) no identifiable inclusions, (B) bright inclusions only, (C) dark inclusions only, (D) bright and dark inclusions. After microCT scanning, grains for $^3$He measurement and Li analysis were picked from the mounts according to this classification. The volume of certain magnetite grains and their respective inclusions was determined in 3D Slicer by segmenting these structures by contrast with manual adjustments in all successive slices and then
interpolating between the segments to form volume models.

## 2.4 Bulk chemistry and Li measurements

Bulk elemental compositions, Li concentrations, and concentrations of α-producing elements (U, Th, Sm) were measured for bulk soil samples, magnetite, and quartz in order to constrain the cosmogenic and radiogenic neutron flux as well as the resulting production of $^3$He via the $^6$Li(n,α)$^3$H(β$^-$)$^3$He pathway and the production of $^3$He from ternary fission. Sample
processing and measurement of elemental composition for bulk samples were performed by Activation Laboratories Ltd. (Ancaster, Ontario, Canada), according to their standard protocols UT-6M and UT-7. Around 10 g of bulk soil samples were pulverized in a ring and puck mill. The resulting powder was fused with sodium peroxide in a zirconium crucible and dissolved with concentrated HNO$_3$ and HCl. Samples of magnetite and quartz/feldspar of around 0.25 g were pulverized using mortar and pestle and then dissolved using HF, HClO$_4$, HNO$_3$, and HCl. Extremely resistant phases, such as zircon,
might not have been completely dissolved by this procedure. A list of 50 elements (including Fe, Ti, P, Li, B, Gd, U, Th, and Sm) was measured on these solutions by ICP-MS using an Agilent 7900. The results were verified by interspersed analyses of certified reference materials (PTM-1a, NIST 696, DTS-2b, OREAS 74a/101a/124/139/247/621/629/680/922, CZN-4, NCS DC73520/86315/86303/86314, CCU-1e, AMIS 0368, DMMAS123) processed in the same way as the samples.

## 2.5 Helium measurements

Aliquots for $^3$He measurement were composed of between 5 and 131 grains of magnetite of 250-1200 µm diameter. Most aliquots were analyzed as is, but some were crushed under ethanol using mortar and pestle prior to analysis to investigate the role of fluid inclusions on $^3$He concentrations. Grinding samples prior to thermal degassing to separate the contributions from fluid inclusions and bulk helium is a standard procedure for $^3$He measurements for phases such as olivine (e.g. Kurz,





1986). In order to prevent the potential adsorption of helium during crushing (Protin et al., 2016), samples were ground in a
mortar and pestle under ethanol (see Cox et al., 2017).

Magnetite aliquots of between 4 mg and 58 mg were weighed and wrapped in tin foil. These samples were degassed at 1300
°C in a double-walled vacuum furnace. The evolved gas was purified by adsorbing certain components on activated charcoal
at liquid nitrogen temperatures and by reacting it with two SAES NP10 getters. The remaining gas was then cryo-focused at
13 K and helium was released into the Helix SFT mass spectrometer at 32 K. Measurement of $^3$He and $^4$He was performed
195   by pulse-counting and on a Faraday detector, respectively.

Sensitivity was calibrated against frequently interspersed standards with known amounts of $^3$He and $^4$He. The reproducibility
of standard measurements was 0.7 % (1σ) over the course of a measurement block of two days. Blanks and re-extracts at
1350 °C were performed before and after sample measurements to ensure full helium extraction and to monitor background
levels. Sample measurements were blank-corrected using full procedural blank analyses, which accounted for between 1 and
23 % of the total signal, with an average of 6 %. Blank-corrections and standard repeatability were taken into account when
calculating the reported analytical uncertainties of $^3$He concentrations. Measured $^3$He concentrations are given in millions of
atoms per gram (Mat g$^{-1}$). All measured $^3$He/$^4$He ratios were normalized to an atmospheric ratio of $1.38 \cdot 10^{-6}$ and are given in
atmospheric units (Ra).

## 3 Results

### 3.1 MicroCT

The internal structure of 3385 magnetite grains with a total mass of 1.7 g was investigated through microCT scans of 20
boxes containing a combined 120 layers of grains. Magnetite displayed a uniform microCT contrast (Fig. 4), while other
phases appeared brighter or darker according to their density and composition relative to magnetite (e.g. Ketcham and
Carlson, 2001). Fractures within grains were also detected, but not used for classification since diffusion of helium in
magnetite at earth-surface conditions is likely not important (see above). Based on the contrast, 52 scanned grains (1.5 %)
were discovered to not be magnetite and were classified as "misidentified". Comparisons with known phases (Fig. 4) show
that very bright inclusions have the same contrast as zircon, and inclusions with a brightness in between those and magnetite
are close to that of apatite. Inclusions that are darker than magnetite have a microCT contrast close to that of quartz and other
silicates.

The 3333 scanned magnetite grains had diameters between 250 and 1500 µm, with most grains being between 300 and 800
µm (Fig. 5a). Overall, 1115 grains (33.5 %) were classified as having no inclusions within the resolution of the scan, and
being suitable for cosmogenic $^3$He dating. There were 417 grains (12.5 %) with bright inclusions, 1046 grains (31.4 %) with
dark inclusions, and 755 grains (22.6 %) with bright and dark inclusions.

Smaller grain size fractions yielded a greater number of grains without inclusions, and the largest fraction of useful grains
(around 50 %) was observed for grain diameters of 200-400 µm (Fig. 5b,c), with a linear decrease for larger grain diameters



to <20 % over 800 µm. This relationship was observed for the overall data as well as for each individual sample. This trend is consistent with inclusions being spatially distributed at random. Grains with diameters between 400 and 800 µm contributed 77 % of the total mass of usable magnetite grains (Fig. 5d).

For several aliquots (17WW-02-Incl1 through -Incl6), bright and dark inclusions were segmented in the microCT imagery, and isosurface renderings of all grains and inclusions were constructed (Figs. 6-8). The volume of magnetite grains and inclusions was determined from these segmented models. Bright inclusions are on average smaller than dark inclusions. Most bright inclusions are elongated in one axis (Figs. 7, 8), which is between 20 and 300 µm in length. Dark inclusions are generally more equant (Figs. 6-8) with diameters generally between 50 and 400 µm. Bright inclusions take up between 0.003 % and 12.6 % of the grain volume, with an average of 0.5 %. Dark inclusions take up 0.03 % to 14.4 % of the grain volume,

with an average of 1.9 %. Dark inclusions are therefore both larger and more numerous than bright inclusions.

### 3.2 Elemental composition

Elemental analysis of ~10 g of bulk soil material yielded Li concentrations of 10-14 ppm, which are constant with depth (see Tab. 1). Lithium is present at a higher concentration (66 ppm) in the clay fraction. Boron was below the detection limit (< 10 ppm) in the bulk soil and around 20 ppm in the clay fraction. Iron is present at ~3 % in the bulk soil and 6 % in the clay

fraction, indicating the presence of detrital iron-oxides, such as magnetite, and pedogenic-iron oxides, which are more concentrated in the clay fraction. Radiogenic elements (U, Th, Sm) were measured at averages around 3 ppm, 36 ppm, and 10 ppm, respectively, in the bulk soil. Major and trace element compositions of the bulk soil did not significantly differ from that of the unweathered fanglomerate (sample 17WW-B1), which represents the parent material of the soil.

The elemental analysis of ~0.25 g of magnetite picked from the magnetic fraction revealed Ti concentrations of 0.26-0.51

mass-%, which allow for a maximum substitution of Ti for Fe of 0.3-0.7 mol-%. Combined with the ATR-FTIR spectra (see Fig. 3). These data show that the extracted phase is relatively pure magnetite without major intergrowth or contribution from other solid-solution endmembers. Trace elements are generally present at lower levels in magnetite than in the bulk soil. Li concentrations are 1.4-2.0 ppm, while U and Th concentrations have averages of 1 ppm and 4 ppm. The P concentrations measured in magnetite allow for the presence of apatite of up to 0.2 % by mass or 0.35 % by volume. An analysis of quartz

and feldspar from the unweathered protolith yielded a Li concentration of 5 ppm. The full results of ICP-MS measurements are reported in the EarthChem database (see Data availability).

### 3.3 Helium concentrations

### 3.3.1 Unscreened magnetite samples

Grains picked at random from the magnetic separate without information about inclusions yielded $^3$He concentrations of 5-

39 Mat g$^{-1}$ (Fig. 9), and $^4$He concentrations of 0.4-4.4 nmol g$^{-1}$ with an average of 1.6 nmol g$^{-1}$ (Tab. 2). Since, on average, around two thirds of scanned grains were found to contain inclusions, it is likely that this unscanned material contains many



bright and dark mineral inclusions. The $^3$He concentrations at every depth (Fig. 9) show large scatter and are up to a factor of four in excess of the expected value based on the known exposure age from $^{10}$Be in quartz (Owen et al., 2014; Hofmann, 2019), as well as the production rate of $^3$He in magnetite and the inherited concentrations calibrated here (see discussion below). The $^3$He depth profile reaches a maximum value of 40 Mat g$^{-1}$ at 20-30 cm and then declines with increasing depth to <10 Mat g$^{-1}$ at 175 cm.

### 3.3.2 Grains with inclusions

Magnetite grains with inclusions, as classified using microCT, also show significant $^3$He excess and the same depth trend as the unscreened magnetite samples (Tab. 3). The $^3$He concentrations of these samples are 4.9-30.6 Mat g$^{-1}$. $^4$He concentrations are 0.1-6.3 nmol g$^{-1}$ with an average of 1.1 nmol g$^{-1}$. Combined with measurements of the volume of inclusions by microCT (see Figs. 6-8), these data show that bright inclusions, such as zircon and apatite, contribute significantly to the $^3$He and $^4$He concentration (Fig. 10) compared to aliquots without inclusions. The contribution of these inclusions is roughly linear with the inclusion volume. Dark inclusions, which are most likely silicates such as quartz, feldspar, and pyroxene, moderately contribute to the $^3$He concentrations and do not increase the $^4$He concentration.

### 3.3.3 Grains without inclusions

Grains without inclusions, as confirmed by microCT, have $^3$He and $^4$He concentrations that are significantly lower than those with inclusions and unscreened magnetite grains (Tab. 4). The $^3$He concentrations of these samples are close to the expected cosmogenic depth profile (Fig. 9) based on the production rate and inherited concentration calibrated (see discussion below). Repeated measurements overlapped within uncertainty. These samples also yielded $^4$He concentrations which are ~0.18 nmol g$^{-1}$ for 12 aliquots, with two aliquots around 1.5 nmol g$^{-1}$.

### 3.3.4 Ground vs. unground aliquots

Grinding grains prior to measurement should release any helium in fluid inclusions (Kurz, 1986). We did not find any significant difference between ground and whole grain $^3$He concentrations of aliquots with and without inclusions (Fig. 11). Aliquots with dark inclusions show no difference in the $^4$He concentrations between ground and whole grains. Ground aliquots with bright inclusions have both higher and lower $^4$He concentrations than unground aliquots.

## 4 Discussion

### 4.1 MicroCT imaging as a screening tool

Data presented here demonstrate that inclusions in magnetite significantly influence the $^3$He and $^4$He concentrations. This presents an analytical challenge that must be addressed since magnetite frequently hosts mineral inclusions (Nadoll et al.,



2014). For example, magnetite and apatite are often closely associated, especially in a hydrothermal setting, such as those producing iron-oxide-apatite ores (e.g. Nadoll et al., 2014). Our microCT scans reveal that most magnetite grains in this study have mineral inclusions, including zircon, apatite, quartz, and other silicates. Previous studies have successfully used microCT screening to avoid [4]He contributions from high-eU inclusions in magnetite for use in (U-Th)/He dating (Cooperdock and Stockli, 2016; Cooperdock et al., 2020, Schwartz et al., 2020). We demonstrate that microCT scanning can

also be used to assess the suitability of magnetite grains for cosmogenic nuclide studies.

We classified inclusions into bright or dark based on the microCT contrast relative to magnetite (Fig. 4). In order to assign mineral phases to these inclusion types, we compared them to microCT images of known phases (see Fig. 4) and analyzed the helium content of magnetite grains with different inclusions. There are several shades of bright inclusions, which likely correspond to apatite and zircon based on comparison of the microCT contrast of these inclusions with known phases.

Magnetite grains with bright inclusions have significantly elevated [4]He concentrations as compared to inclusion-free grains (see Fig. 10). This implies that bright inclusions have high effective uranium (eU) concentrations, by which they have accumulated radiogenic [4]He since the closure of the system. This is consistent with apatite and zircon crystals, which typically have hundreds to thousands of ppm of eU (e.g. Farley, 2002). In addition, the size (50-300 µm) and prismatic habit of bright inclusions are similar to that of apatite and zircon.

Dark inclusions, by comparison, contribute much less to the [4]He concentration than bright inclusions (see Figs. 9,10), which is evidence for a relatively lower eU content. This is consistent with the typically low U, Th, and Sm concentrations of quartz and feldspar (e.g. Vandenberghe et al., 2008). The microCT contrast of quartz, feldspar, and other silicates is also similar to that of dark inclusions observed in magnetite. The typical diameter (100-400 µm) and mostly equant habit of these dark inclusions is also consistent with them being quartz and feldspar.

Dark (less dense) inclusions might also be fluid inclusions. However, the [3]He concentrations of ground and unground aliquots were found to be the same within uncertainty (Fig. 11). The variability in [4]He concentrations between ground and unground aliquots is most likely not an effect of the grinding, but due to the stochastic nature of the number of inclusions in any given magnetite grain. The variability of the [4]He concentrations between ground and unground aliquots mirrors that of the aliquots with bright inclusions and of unscanned magnetite (see Fig. 9). Since grinding does not release any measurable

amounts of helium, we conclude that fluid inclusions are not a significant contributor to the overall [3]He concentration, thus the dark inclusions seen on microCT imaging in these samples are likely not fluid inclusions.

Since both bright and dark inclusions contribute significant [3]He (and [4]He) compared to inclusion-free magnetite (see Fig. 9), we selected aliquots consisting only of grains without inclusions, as detectable by microCT. Measured [3]He in these aliquots is likely the result of mainly cosmogenic production in magnetite, with some production of [3]He from Li and ejection into or

implantation from neighboring phases (see discussion below).



## 4.2 Production of $^3$He in magnetite

In order to translate measured $^3$He concentrations into an exposure age and calibrate the production rate in magnetite, the production $^3$He from various sources needs to be considered. Production of $^3$He by cosmogenic high-energy neutrons is likely the main pathway for most of the $^3$He measured in magnetite grains without inclusions. Determining this component

allows the calibration of the cosmogenic $^3$He production rate in magnetite. Due to low (<10 ppm) B concentrations, production of nucleogenic $^3$He from $^{10}$B is negligible (Lal, 1987). Relatively low U concentrations in the magnetite (~1 ppm) and the matrix (~3 ppm) lead to a negligible contribution of $^3$He from fission (Farley et al., 2006). We correct the measured $^3$He concentrations in magnetite aliquots without inclusions for $^3$He produced via the $^6$Li(n,α)$^3$H(β$^-$)$^3$He pathway (Andrews and Kay, 1982) in order to yield solely the cosmogenic component. Since the mean free path length of thermal neutrons is

around 50 cm in granite (Lal, 1987), the thermal neutron flux is independent of the mineral-scale U and Th concentrations (Farley et al., 2006; Amidon et al., 2008) and is dependent on the U and Th concentrations of the bulk soil. Production of $^3$He from $^6$Li is therefore mainly controlled by the local Li concentration within the magnetite grain as well as in an area around it from which $^3$H could be implanted into the magnetite grain.

To determine the contribution of nucleogenic $^3$He, which has accumulated since the He closure of the magnetite system, we

estimated cooling ages from measured parameters for each aliquot. Since these are detrital magnetite grains, it is unclear whether all grains within an aliquot share the same source, and should have the same cooling age, or represent multiple sources. We combined the measured $^4$He concentrations of 0.1-1.6 nmol g$^{-1}$ in individual magnetite aliquots without inclusions with average magnetite U and Th concentrations of 1.1 ppm and 4 ppm to calculate (U-Th)/He ages of 9-143 Ma, with most ages being 15±5 Ma and two ages at 122 Ma and 143 Ma. Since these aliquots do not contain any inclusions, we

interpret these ages as the average cooling age of each aliquot, which was calculated solely to enable the nucleogenic $^3$He correction of individual aliquots.

For comparison, apatite (U-Th)/He cooling ages (AHe) of basement rocks in the source region of the Whitewater fan deposits (Fosdick and Blisniuk, 2018) are 5-13 Ma in the southern part and 53-56 Ma in the northern part (Spotila et al., 2001). Since the magnetite (U-Th)/He system (Blackburn et al., 2007) has a much higher closure temperature (~250 °C) than

apatite (~75 °C), these ages are likely a lower bound for the cooling ages of magnetite from the same source rocks. Detrital U-Pb zircon ages (closure temperature ~900 °C) of the Whitewater River show major contributions from rocks of the 72-80 Ma Sierra Nevada batholith, with smaller contributions from 155-180 Ma batholithic rocks, a 215 Ma mega porphyry, a 1.4 Ga leucogranite, and a 1.7 Ga augen gneiss (Fosdick and Blisniuk, 2018). Our estimated magnetite (U-Th)/He ages are consistent with magnetite cooling through its He closure temperature between the AHe cooling ages and the zircon U-Pb

ages for the possible sediment source regions. Younger cooling ages likely represent a contribution from the southern part of the source region of the Whitewater fan deposits, whereas older cooling ages are likely from the northern part.

The radiogenic thermal neutron rate produced by 3 ppm U and 36 ppm Th (as measured in the fanglomerate) was estimated to be 1.9 n g$^{-1}$ a$^{-1}$ and 27 n g$^{-1}$ a$^{-1}$ (see Andrews and Kay, 1982). The combined RTN production rate of 32.5 n g$^{-1}$ a$^{-1}$ gives rise





to a neutron flux of around 1800 n $^{-1}$ cm$^{-2}$ a$^{-1}$, which leads to a specific $^3$He production rate of 0.011 at g$^{-1}$ a$^{-1}$ per 1 ppm Li

according to the methodology of Lal (1987). If the distribution of α-producing elements is not homogeneous, this is likely to be a maximum estimate (Farley et al., 2006). For 1.6 ppm Li in magnetite and 12 ppm in the matrix, this yields nucleogenic $^3$He production rates of 0.018 at g$^{-1}$ a$^{-1}$ and 0.13 at g$^{-1}$ a$^{-1}$. Using the estimated cooling age for each sample yielded nucleogenic $^3$He concentrations of 1 Mat g$^{-1}$ for ~15 Ma, and 7 Mat g$^{-1}$ for ~130 Ma. This predicted nucleogenic contribution was subtracted from the bulk $^3$He concentration.

A second contribution of $^3$He from Li is derived from reactions with cosmogenic thermal neutrons accrued over the exposure duration of the magnetite grains (Dunai et al., 2007). Assuming a typical granitic composition (e.g. Amidon et al., 2008) with 2 % water content and a density of 1.9 g cm$^{-3}$ combined with measured values for Li, B, Na, Mg, Al, P, Fe, Ti, Si, K, and Gd (Tab. 1 and EarthChem database, see Data availability) we calculated a combined thermal and epithermal cosmogenic neutron flux of 72100 n g$^{-1}$ a$^{-1}$ using CHLOE (Phillips and Plummer, 1996; Phillips et al., 2001). This is the neutron flux at

the surface scaled for latitude, longitude, and elevation. This CTN flux yields a surficial $^3$He production rate of 0.442 at g$^{-1}$ a$^{-1}$ per 1 ppm Li. Average Li concentrations in Whitewater magnetite and the bulk soil of 1.6 ppm and 12 ppm yield surficial $^3$He production rates of 0.71 at g$^{-1}$ a$^{-1}$ and 5.3 at g$^{-1}$ a$^{-1}$. The depth profile of CTN production was modeled using CHLOE, with the above parameters and a fast-neutron attenuation length of 160 g cm$^{-2}$. The maximum CTN production rates are 1.2 at g$^{-1}$ a$^{-1}$ and 9.0 at g$^{-1}$ a$^{-1}$ at a depth of 22 cm. Over the exposure age of 53.5 ka, as derived from $^{10}$Be and $^{26}$Al depth profiles,

the CTN contribution to the $^3$He concentration is between 0.07 and 0.09 Mat g$^{-1}$, depending on the depth. This component was also subtracted from the measured bulk $^3$He concentrations.

Accounting for Li-produced $^3$He decreased the overall scatter of the depth profile (Fig. 12). Replicate measurements of samples from the same depth have $^3$He concentrations that overlap within uncertainty. Remaining deviations of measured $^3$He concentrations from an exponential depth profile can be explained by either an inhomogeneity in Li concentration

between magnetite aliquots, inclusions that were not detected by microCT, or exposure prior to the deposition of the fanglomerate.

### 4.3 Calibration of the $^3$He production rate in magnetite

The magnetite $^3$He production rate was calibrated by comparing the depth profile of grains without inclusions (Fig. 12) to a known exposure age of 53.5±2.2 ka from a depth profile of $^{10}$Be and $^{26}$Al in quartz (Hofmann, 2019) extracted from the same

sample material. While this exposure age was calculated assuming a zero erosion rate, we observed evidence for surface erosion, predominately by deflation, which suggests that the measured exposure age is likely a minimum estimate of the true exposure age. However, since both the quartz and magnetite grains come from the same grusified clasts, which were still coherent when sampled, they share a common thermal and exposure history. The coarse-grained texture of the protolith, which is still readily apparent, also implies no vertical movement of these clasts after the deposition of the fanglomerate. The

concentrations and production rates of $^{10}$Be in quartz and $^3$He in magnetite are therefore directly comparable.





Magnetite $^3$He concentrations were corrected for their nucleogenic and CTN-produced components using the procedure outlined above. The corrected $^3$He concentrations are close to an exponential trend with depth (Fig. 12). One obvious outlier at 60 cm depth was excluded for the production rate calibration. We used a Monte-Carlo forward model to compute a best-fit exponential depth profile assuming a bulk soil density of $\rho$ = 1.9±0.1 g cm$^{-3}$ and an effective attenuation length $\Lambda_{eff}$ = 160±5

g cm$^{-2}$ (estimate for latitude, see Dunai, 2001). The depth-constant inherited concentration $N_i$ and the cosmogenic production rate P were optimized simultaneously, and the 95 % uncertainties of these parameters were calculated. This model yielded an inherited concentration of 1.7±0.6 at g$^{-1}$ (2$\sigma$) and an apparent local surface production rate of 158±18 at g$^{-1}$ a$^{-1}$ (2$\sigma$). This local production rate was scaled to sea-level and high latitude (SLHL) using a time-constant scaling factor of 1.358 (Lal, 1991). The SLHL $^3$He production rate determined for the Whitewater site is 116±13 at g$^{-1}$ a$^{-1}$ (2$\sigma$).

Our magnetite SLHL production rate is higher than some previous estimates from element-specific production rates of 66 at g$^{-1}$ a$^{-1}$ (Masarik and Reedy, 1996) and calibrations of 69-77 at g$^{-1}$ a$^{-1}$ (Bryce and Farley, 2002). It is, however, within uncertainty of the modeled (122 at g$^{-1}$ a$^{-1}$) and calibrated (120±12 at g$^{-1}$ a$^{-1}$) production rate of Kober et al. (2005) for Fe-Ti-oxides. The relatively pure magnetite used in this study is very similar in elemental composition and density to the Fe-Ti-oxides used by Kober et al. (2005), which makes it possible to compare our results to these production rate estimates.

With the above SLHL $^3$He production rate in magnetite and a $^{10}$Be production rate at SLHL of 3.92 at g$^{-1}$ a$^{-1}$ (Borchers et al., 2016), the $^3$He/$^{10}$Be production ratio is 29.59±4.6 (2$\sigma$). A direct comparison of the $^3$He and $^{10}$Be concentrations, which were corrected for decay according to Blard et al. (2013), from the same depths (Fig. 13) leads to a $^3$He/$^{10}$Be production ratio of 28.3±4.8 (2$\sigma$). These estimates are within uncertainty of previously measured ratios (Kober et al., 2005) and are in the same range as those of most common mineral phases at 500 m elevation (Blard et al., 2013).

**4.4 Source of excess $^3$He from inclusions**

We observed high $^3$He concentrations in magnetite aliquots with inclusions compared to those without inclusions. The cosmogenic $^3$He production rates of most silicates are between 100 and 120 at g$^{-1}$ a$^{-1}$ (e.g. Goehring et al., 2010), similar to the magnetite $^3$He production rate determined here. Therefore, inclusions such as those observed here should not significantly alter the cosmogenic $^3$He production within the grain. Further, we have observed no $^3$He contribution from fluid

inclusions, hence the increase in $^3$He concentration is attributable to mineral inclusions that have been present within magnetite since the growth of the grains. The magnetite grains and the inclusions have consequently experienced the same temperature conditions and exposure to cosmic rays. Any excess $^3$He as a result of the presence of mineral inclusions is likely produced by non-cosmogenic processes.

One possible source of excess $^3$He is the production of $^3$He due to ternary fission of U (Vorobiev et al., 1969; Halpern,

1971), with a known production rate of 2·10$^{-5}$ at g$^{-1}$ a$^{-1}$ per ppm U (Farley et al., 2006). Zircons typically have hundreds of ppm of U, and in some cases >1000 ppm (e.g. Reiners, 2005). A large part of this radiogenic $^3$He might be ejected from the inclusion and implanted into the magnetite grain due to the large kinetic energy of this process (Farley et al., 2006).



The highest $^4$He concentration measured here is around 35 times higher than the average $^4$He concentration in magnetite without inclusions. Assuming that this is radiogenic $^4$He due to decay of U and Th contributed by zircons, the average eU of

these zircons is around 10000 ppm. If zircons are present at 2 % of the total volume of the aliquot, which is similar to what was observed by microCT for bright inclusions, these inclusions would contribute around 18 Mat g$^{-1}$ to the total $^3$He concentration of 22.4 Mat g$^{-1}$ for this aliquot. This shows that over sufficiently large cooling ages a significant amount of radiogenic $^3$He can be contributed by high-eU inclusions.

Another possible contribution of $^3$He is nucleogenic and CTN production on $^6$Li. Magnetite has been observed in this study

to have relatively low Li concentrations (~1.6 ppm). While zircons are typically low in Li, many other common mineral phases have Li concentrations of tens or hundreds of ppm (Amidon et al., 2008; 2009). Quartz, feldspar, and apatite can have tens, pyroxene and amphibole hundreds, and mica thousands or tens of thousands of ppm Li (Amidon et al., 2009). Using the radiogenic neutron fluxes computed above, silicate inclusions comprising 5 % of the total volume with 100 ppm Li can contribute 8 Mat g$^{-1}$ of nucleogenic $^3$He over a 140 Ma cooling age. This component is independent of depth.

Production of $^3$He from Li due to cosmogenic thermal neutron flux is depth-dependent. The unscreened magnetite aliquots and aliquots with known inclusions display considerable scatter, but there is general decrease of $^3$He concentration with depth (see Fig. 9b,c). This decrease is greater than the expected variation in cosmogenic production, so part of this trend can be attributed to a decrease in CTN production with depth. The cosmogenic neutron flux is highest around 22 cm depth below the surface. At this depth, the CTN production from a quartz inclusion of 5 % by volume and 100 ppm Li would be around 5

% of the cosmogenic production rate in magnetite. Pyroxene, amphibole, or mica inclusions with Li concentrations conceivably >>100 ppm might contribute an even greater amount of nucleogenic and CTN $^3$He.

Any randomly chosen aliquot of magnetite is likely to have inclusions of various minerals of several volume-%. These inclusions contribute $^3$He from combined radiogenic, nucleogenic, and CTN production due to their increased U and Li content relative to magnetite, which leads to $^3$He concentrations that are in excess of the expected cosmogenic production, as

observed in this study (up to a factor of 4). The effect of mineral inclusions might be even greater in other sites where rocks have older cooling ages. This highlights the importance of avoiding mineral inclusions in the target phase when determining $^3$He concentrations for cosmogenic exposure studies.

Some types of mineral inclusions, which have U and Li concentrations comparable to or below those of the host phase, might not present a problem for determining $^3$He concentrations. Gayer et al. (2004), for instance, found no difference in the

$^3$He concentration between garnet aliquots with many ilmenite inclusions and those with fewer inclusions. Ilmenite has neither high U nor Li concentrations, therefore it would not significantly contribute to the overall $^3$He concentration. However, without sufficient characterization any mineral inclusion can be a potential concern and should be avoided.

## 4.5 Applicability to opaque phases

We show that microCT scanning of magnetite grains significantly improves the quality and reproducibility of $^3$He

measurements, making their use in cosmogenic exposure dating more robust and reliable. The utility of magnetite for



deriving watershed-averaged erosion rates using [36]Cl has already been demonstrated, due to its resistance to erosion similar to quartz (Moore and Granger, 2019b). The microCT screening approach makes it feasible to also use [3]He in magnetite for the same purpose. Due to its relatively high helium-retentivity, magnetite is resistant to thermal resetting of cosmogenic [3]He (Blackburn et al., 2007), and could therefore also be employed for paleo-exposure studies similar to those using goethite
(Hofmann et al., 2017) and olivine (Balbas and Farley, 2020).

The screening approach using microCT presented in this study might also be applied to other opaque helium-retentive phases, such as pyroxene, biotite, and hornblende, which are already in use as target phases (e.g. Amidon and Farley, 2011; 2012). Due to the large mass requirements this method is likely impractical for routine use with radioactive cosmogenic nuclides, but might be utilized for studies using magnetite and other opaque phases with >1 Ma exposure ages for which a
few grams of material are sufficient (e.g. Matsumura et al., 2014).

## 5 Conclusions

We find that mineral inclusions including quartz, feldspar, apatite, and zircon contribute significant amounts of [3]He to magnetite grains, which, in the case of samples from the Whitewater site, can lead to an excess of a factor of four above the cosmogenically produced amount of [3]He. These elevated concentrations are caused by radiogenic [3]He from high-eU
inclusions such as apatite and zircon, and implantation of [3]He from thermal neutron production on inclusions with higher Li concentration as magnetite, such as quartz and other silicates. We did not find any significant amount of [3]He derived from fluid inclusions.

Contribution of [3]He from mineral inclusions can be prevented by screening magnetic separates using microCT and selecting only those grains without inclusions for analysis. Most grains deemed to be suitable in this study were found to have grain
diameters of 400-800 µm. Knowledge of the U and Li concentrations of the magnetite grains are important to assess the nucleogenic, CTN, and radiogenic components of the [3]He in the sample. Correcting the measured [3]He concentrations for non-cosmogenic production of [3]He improved the scatter of the depth profile.

We calibrated the production rate of [3]He in magnetite by comparing corrected [3]He concentrations to an existing depth profile of [10]Be and [26]Al in quartz. This yielded a [3]He production rate in magnetite of $116\pm13$ at $g^{-1}$ $a^{-1}$ (2σ) at sea level and high
latitude, which is comparable to previous calibrations (Kober et al., 2005).

By screening magnetite separates using microCT to select only inclusion-free grains and accounting for non-cosmogenic production, [3]He in magnetite can be used as a robust tool for in-situ and detrital cosmic-ray exposure studies. This technique might also help to improve the quality of cosmogenic [3]He measurements of other opaque phases.



## 6 Data availability

All elemental analyses and helium isotope data are available at https://doi.org/10.26022/IEDA/111932. MicroCT scans have been uploaded to http://digitalrocksportal.org and will be cited with a DOI in the final version of this manuscript after review.

## 7 Author contributions

FH, EC, AJW, and KAF conceptualized the study and acquired funding. FH, DH, and KS carried out mineral separation,
sample processing, and data analysis. FH, EC, AJW, and KAF contributed to the interpretation of the data. The manuscript and figures prepared by FH were edited and reviewed by all co-authors.

## 8 Competing interests

The authors declare that they have no conflict of interest.

## 9 Acknowledgements

We thank Tautis Skorka at the Molecular Imaging Center at the University of Southern California for microCT scanning. Jonathan Treffkorn is thanked for performing helium mass spectrometry at Caltech.

## 10 Financial support

Funding was provided by Southern California Earthquake Center 2020 Award #20146 awarded to Kenneth Farley, Emily Cooperdock, Joshua West, and Florian Hofmann. Student contributions at Ludwig-Maximilians-Universität München were
funded by Studi_forscht@GEO grant S20-F17 awarded to Florian Hofmann.

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



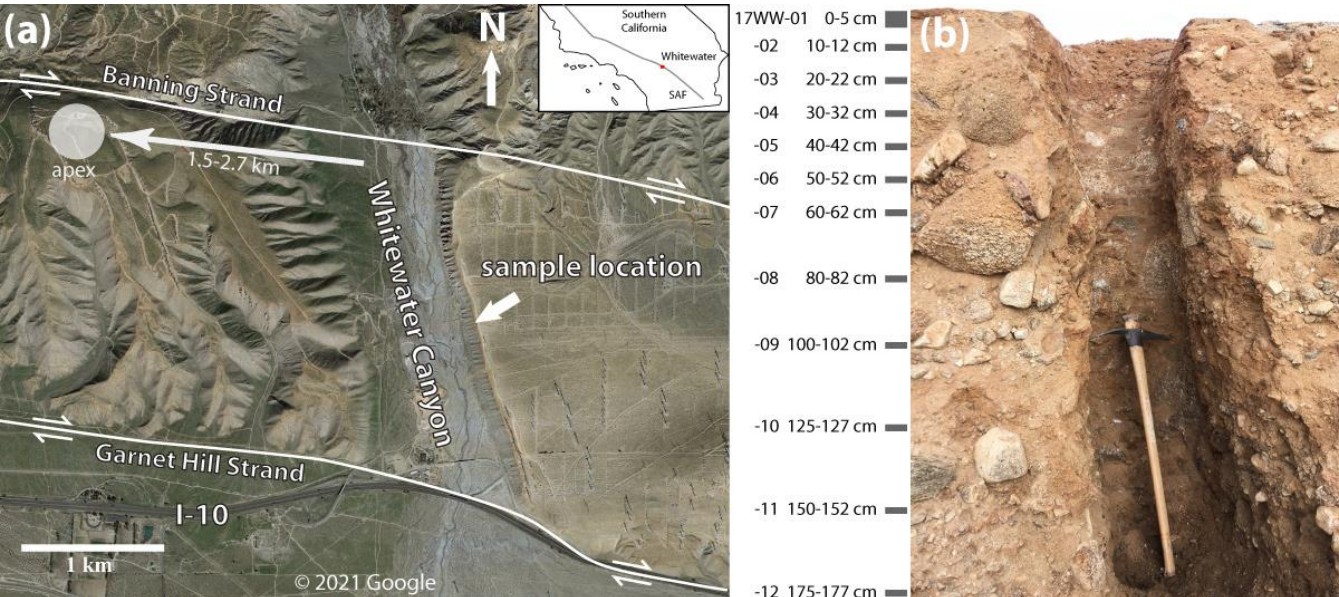

**Figure 1: (a) Map of the offset alluvial fan complex at Whitewater, California, USA., which was offset from the mouth of the**
**Whitewater River by the Banning Strand of the San Andreas fault. The current location of the fan apex (Huerta, 2017) as well as**
**the offset vector are shown. (b) The relict soil capping the terrace was sampled from the surface to 1.77 m depth. Sample-IDs are**
**shown next to depth intervals. Magnetite was separated from these samples for microCT scanning and $^3$He measurement. Aerial**
**imagery taken from © Google Earth, fault traces after Kendrick et al. (2015), offset alluvial fan apex from Huerta (2017).**


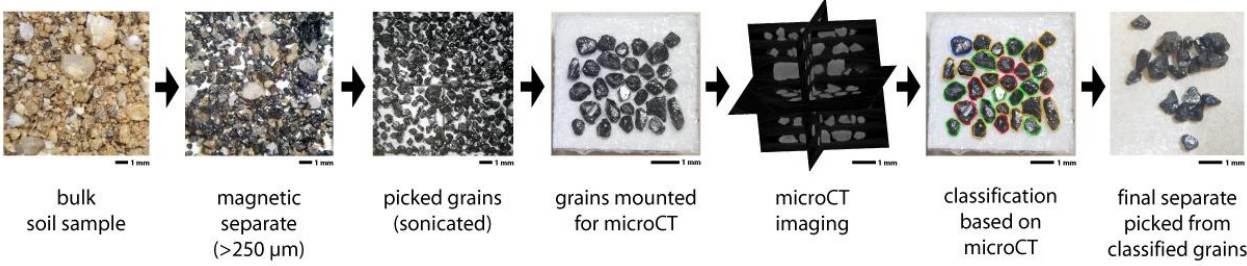

**Figure 2: Sample processing workflow from bulk soil samples to final separate picked from microCT-analyzed grains. The**
**microCT mounts are approximately 5 mm in width. Individual grains were classified as having no inclusions or containing**
**bright/dark/bright and dark inclusions according to microCT analysis (see Fig. 4).**


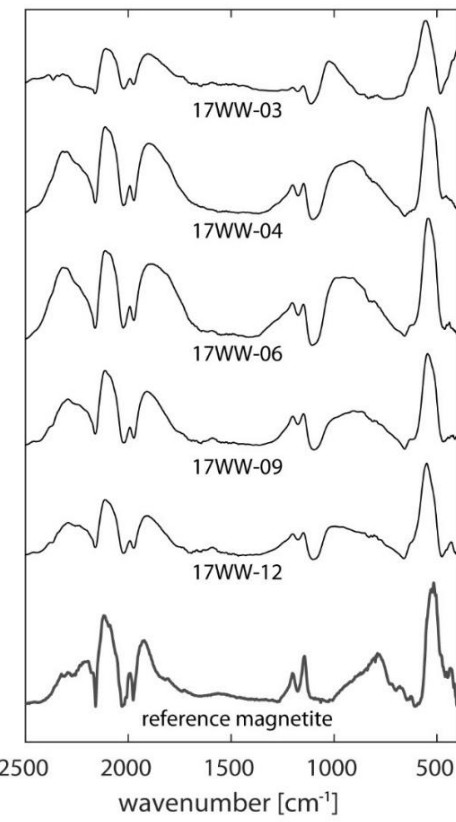

**Figure 3: Baseline-corrected Attenuated Total Reflection Fourier-Transform Infrared (ATR-FTIR) spectra of powdered samples, showing a known magnetite sample (bottom) and picked magnetic separates of five different depths. These spectra demonstrate that the sample material is composed of magnetite without any detectable contributions from other phases. This method is not sensitive to substitution of Fe by other elements.**



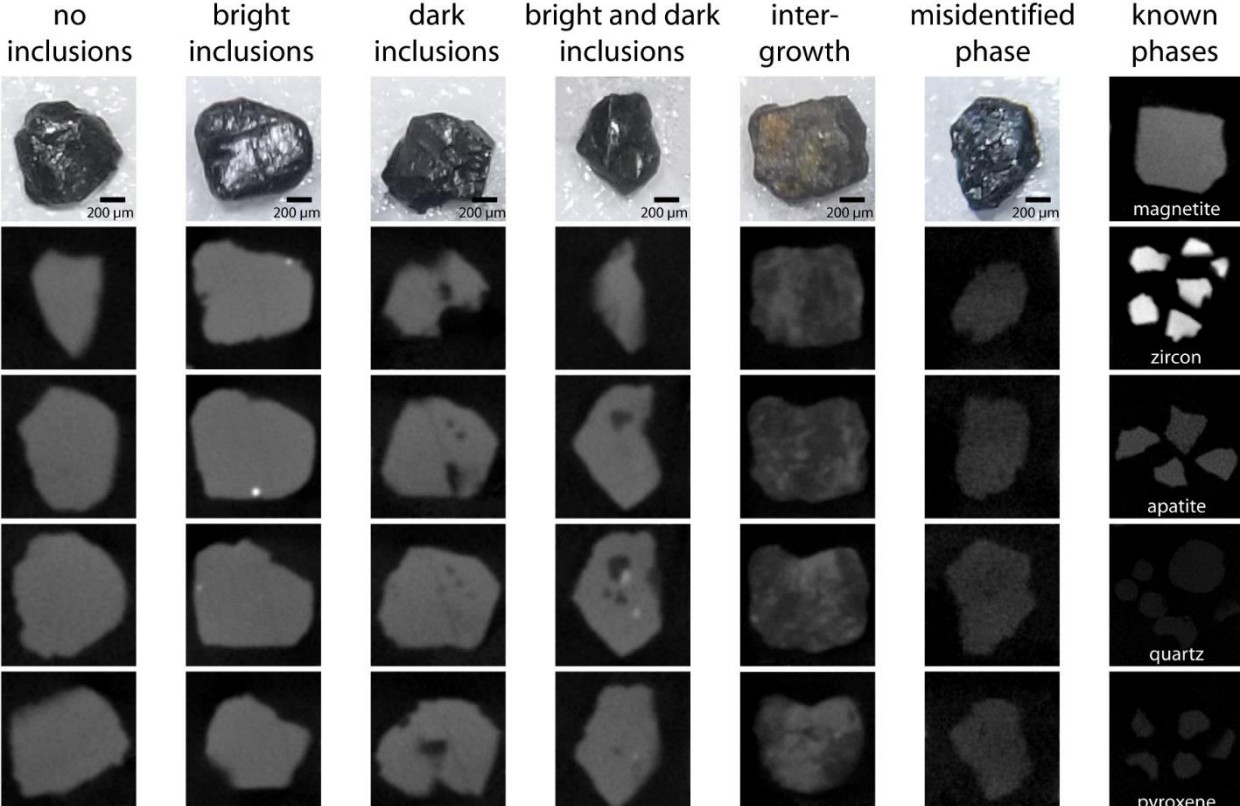

**Figure 4: Examples of textures and inclusions in magnetite grains detected with microCT. Light micrographs and four horizontal microCT slices through the same grain are given in each column. All microCT scans were acquired with the same scan parameters, and images are given at the same contrast. Reference microCT images for known phases are given for comparison (magnetite as confirmed by FTIR, SLI 3 zircon, Durango apatite, and CRONUS quartz and pyroxene standards). Bright inclusions represent zircon and apatite, whereas inclusions darker than magnetite are most likely silicates. Intergrowth/substitution structures and other phases mistakenly identified as magnetite during sample picking under a light microscope were also detected.**



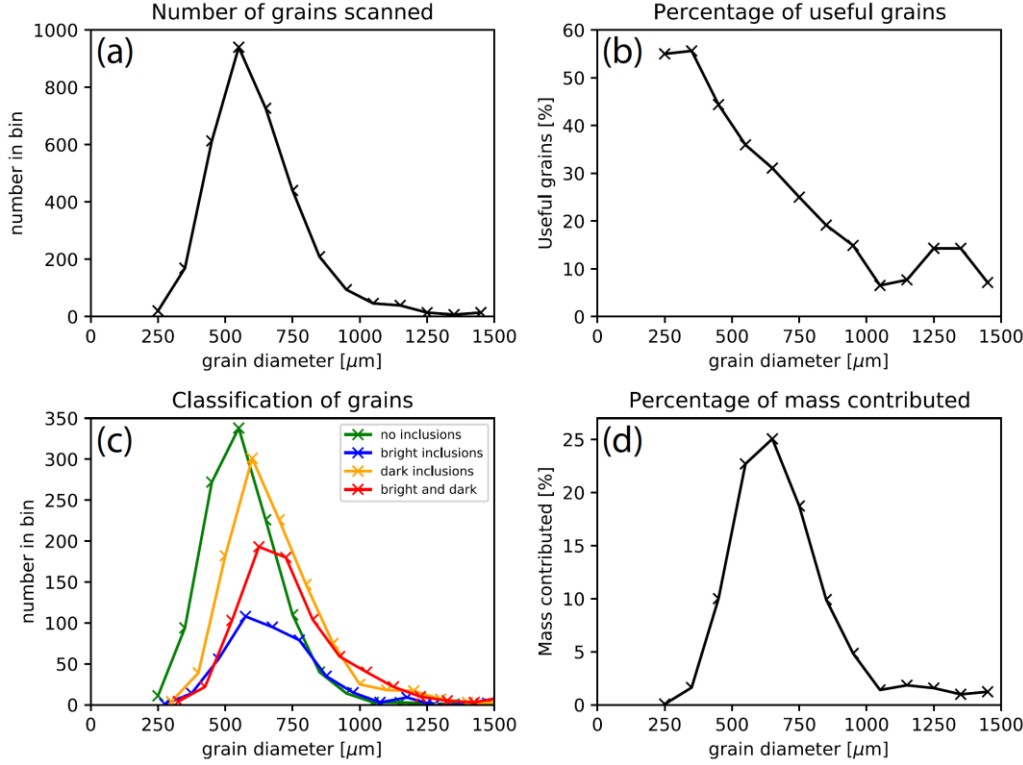

**Figure 5: Results of the classification of grains according to inclusions identified by microCT imaging, showing that the most useful (inclusion free) grain size range for analysis was 400-800 µm. All plots are histograms with grain size bins of 100 µm width. (a) The total number of grains which were scanned, presented by grain size. (b) The percentage of grains for each grain size bin**
**which were found to be suitable (i.e. without inclusions). (c) Classification of grains into no inclusions, bright inclusions, dark inclusions, bright and dark inclusions. (d) The percentage of the total mass of grains used for analysis in each grain size bin. Mass was estimated based on the grain diameter and the mean density of magnetite.**



**Figure 6: Segmented volume renderings of magnetite grains with dark inclusions (dark blue) constructed from microCT data. These are all of the grains from aliquot 17WW-02-Incl2 (only dark inclusions), for helium data see Tab. 3.**





**Figure 7: Segmented volume renderings of magnetite grains with bright (turquoise) and dark inclusions (dark blue) constructed from microCT data. These are all of the grains from aliquots 17WW-02-Incl1 (only bright inclusions) and -Incl3 (bright and dark inclusion), for helium data see Tab. 3.**



**Figure 8: Segmented volume renderings of magnetite grains with many bright (turquoise) and dark inclusions (dark blue) constructed from microCT data. These are all of the grains from aliquots 17WW-02-Incl4 (many bright inclusions), -Incl5 (many dark inclusions), and -Incl6 (many bright and dark inclusion), for helium data see Tab. 3.**





**Figure 9: Depth profiles of measured ³He (a-c) and ⁴He (d-f) concentrations from unscreened magnetite grains without information about inclusions (black), samples with bright (turquoise), dark (blue), as well as bright and dark inclusions (red), and samples without inclusions (green). All uncertainties shown are at the 1σ level and overlapping data points have been slightly vertically offset for clarity. The expected cosmogenic ³He depth profile based on the known exposure age and the average ⁴He concentration of aliquots without inclusions are shown as gray lines. Samples without inclusions have ³He concentrations close to the predicted depth profile and all but two have low, nearly constant ⁴He concentrations (around 0.18 nmol g⁻¹). Grains with inclusions have significantly higher ³He and ⁴He concentrations, showing the effects of radiogenic and thermal neutron produced ³He added by these inclusions. The ³He concentration decreases with depth even in grains with inclusions, indicating a cosmogenic thermal neutron component to nucleogenic production.**

**Figure 10: The effect of bright (turquoise) and dark (dark blue) inclusions on the ³He and ⁴He concentration of magnetite grains compared to aliquots without inclusions (green). Bright inclusions (apatite, zircon) contribute large amounts of ³He and ⁴He, whereas dark inclusions (silicates) contribute little ⁴He and moderate amounts of ³He. Lines show general trends and are not quantitative.**



**Figure 11: Comparison of aliquots of unground (black) and ground (red) magnetite aliquots with 2σ uncertainties. The $^3$He concentrations of unground and ground aliquots are identical within the 95 % uncertainty bounds. Sample 17WW-01 was taken at the surface and 17WW-09 and -11 were taken at 1 m and 1.5 m depth.**






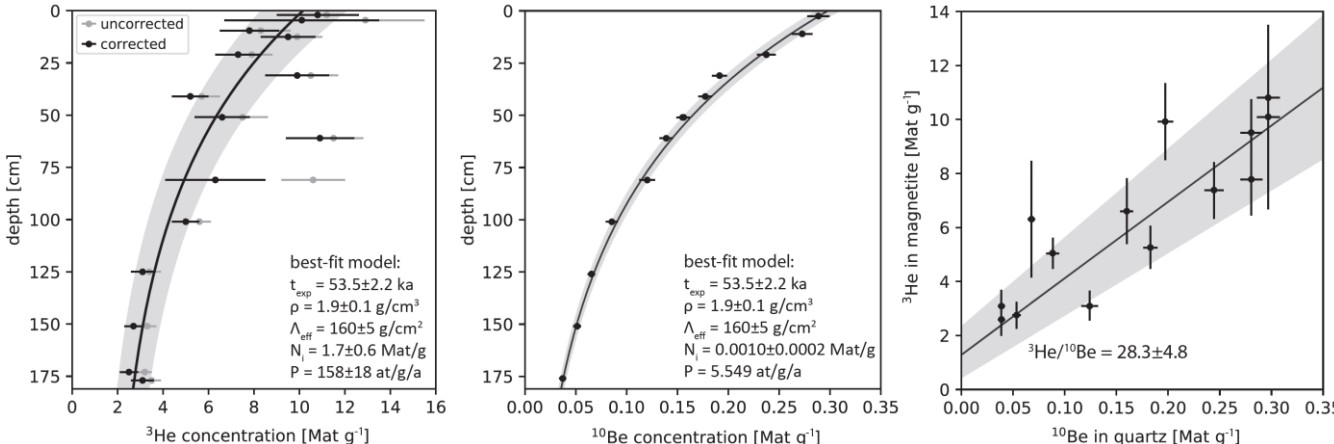

**Figure 12: (a) Depth profile of measured $^3$He concentrations in magnetite without inclusions corrected for nucleogenic and CTN-produced $^3$He with 2σ uncertainties. Uncorrected data are shown in gray. The solid line and shaded area show the best-fit model with 2σ uncertainty using a known exposure age $t_{exp}$, bulk density $ρ$, and effective attenuation length $Λ_{eff}$. The model was optimized for the inherited concentration $N_i$ and the local surface production rate $P$. Replicate analyses at the same depth are shown with a**

**slight vertical offset for clarity. (b) Depth profile of $^{10}$Be in measured in quartz (Hofmann, 2019) extracted from the same samples as the magnetite studied here. The best-fit model uses the same parameters as in (a), but was optimized for the exposure age and inherited concentration, with a scaled surface $^{10}$Be production rate from the CRONUS-Earth online calculator (Balco et al., 2008). This measurement establishes the surface exposure age used to calibrate the production rate in magnetite. (c) Determination of the $^3$He/$^{10}$Be production ratio using data from the from depth profiles. $^{10}$Be concentrations were corrected for decay according to**

**Blard et al. (2013).**





**Table 1: Results of elemental analyses of bulk soil samples and mineral separates by ICP-MS. Only the most relevant parameters**

**for determining neutron fluxes and $^3$He production rates are shown; full results can be found in the EarthChem database.**

| Sample | depth | material | mass | Fe | Ti | P | Li | B | Gd | U | Th | Sm |
|---|---|---|---|---|---|---|---|---|---|---|---|---|
| | [cm] | | [g] | [%] | [%] | [ppm] | [ppm] | [ppm] | [ppm] | [ppm] | [ppm] | [ppm] |
| Bulk chemistry | | | | | | | | | | | | |
| 17WW-02 | 10-12 | bulk soil | 10.7 | 2.98 | 0.29 | | 11 | < 10 | 4.7 | 2.6 | 18.8 | 6.4 |
| 17WW-06A | 50-52 | bulk soil | 10.3 | 3.72 | 0.35 | | 14 | < 10 | 11.4 | 4.5 | 72.8 | 14.9 |
| 17WW-06B | 50-52 | bulk soil | 10.0 | 3.52 | 0.31 | | 12 | < 10 | 8.8 | 3.0 | 47.9 | 12.0 |
| 17WW-08 | 80-82 | bulk soil | 10.4 | 3.79 | 0.34 | | 12 | < 10 | 4.8 | 2.6 | 24.0 | 6.5 |
| 17WW-10 | 125-127 | bulk soil | 10.1 | 3.21 | 0.31 | | 11 | < 10 | 6.5 | 2.3 | 29.4 | 7.8 |
| 17WW-12 | 175-177 | bulk soil | 10.5 | 3.32 | 0.29 | | 10 | < 10 | 5.2 | 3.3 | 19.4 | 5.8 |
| 17WW-08c | 80-82 | clay fraction | 8.7 | 6.32 | 0.69 | | 66 | 20 | 12.0 | 4.0 | 42.3 | 13.8 |
| 17WW-B1 | ~500 | fanglomerate | 10.5 | 4.07 | 0.37 | | 12 | < 10 | 7.2 | 1.9 | 33.3 | 10.0 |
| Mineral-specific | | | | | | | | | | | | |
| 17WW-B1q | ~500 | quartz/feldspar | 0.261 | 0.14 | 0.02 | 130 | 5.0 | | | 1.3 | 3.02 | |
| 17WW-02 | 10-12 | magnetite | 0.279 | 63.80 | 0.40 | 140 | 1.4 | | | 0.8 | 2.29 | |
| 17WW-08 | 80-82 | magnetite | 0.265 | 66.10 | 0.26 | 80 | 1.4 | | | 1.4 | 4.48 | |
| 17WW-12 | 175-177 | magnetite | 0.263 | 66.40 | 0.51 | 320 | 2.1 | | | 1.2 | 4.29 | |






**Table 2: Measurements of $^3$He and $^4$He concentrations in bulk magnetite aliquots, which were not scanned using microCT. They likely contain many grains with inclusions.**

| Sample 17WW- | depth [cm] | mass [mg] | $^3$He conc. [Mat g$^{-1}$] | 1σ [Mat g$^{-1}$] | $^4$He conc. [nmol g$^{-1}$] | 1σ [nmol g$^{-1}$] | $^3$He/$^4$He [Ra] | 1σ [Ra] |
|---|---|---|---|---|---|---|---|---|
| 01-magA | 0-5 | 49.90 | 17.6 | 1.3 | 0.745 | 0.016 | 0.0284 | 0.0004 |
| 02-magA | 10-12 | 98.10 | 14.7 | 0.9 | 1.055 | 0.022 | 0.0168 | 0.0002 |
| 03-magA | 20-22 | 17.22 | 16.1 | 1.4 | 1.319 | 0.027 | 0.0147 | 0.0004 |
| 03-magB | 20-22 | 19.55 | 28.3 | 2.1 | 0.998 | 0.021 | 0.0341 | 0.0006 |
| 03-magC | 20-22 | 20.28 | 8.1 | 1.0 | 0.966 | 0.020 | 0.0100 | 0.0006 |
| 03-magE | 20-22 | 15.57 | 11.8 | 1.1 | 1.094 | 0.022 | 0.0130 | 0.0004 |
| 03-magF | 20-22 | 14.56 | 17.2 | 1.5 | 1.018 | 0.021 | 0.0203 | 0.0005 |
| 04-magA | 30-32 | 29.35 | 16.0 | 1.2 | 3.634 | 0.073 | 0.0053 | 0.0001 |
| 04-magC | 30-32 | 19.89 | 18.0 | 1.4 | 0.712 | 0.015 | 0.0305 | 0.0006 |
| 04-magD | 30-32 | 25.70 | 39.0 | 2.5 | 1.000 | 0.021 | 0.0469 | 0.0006 |
| 05-magA | 40-42 | 21.10 | 14.6 | 1.2 | 0.981 | 0.020 | 0.0179 | 0.0004 |
| 06-magA | 50-52 | 19.16 | 26.1 | 2.0 | 1.931 | 0.039 | 0.0163 | 0.0003 |
| 06-magB | 50-52 | 19.37 | 11.6 | 1.2 | 4.361 | 0.087 | 0.0032 | 0.0001 |
| 06-magC | 50-52 | 18.98 | 9.5 | 0.9 | 4.329 | 0.087 | 0.0026 | 0.0001 |
| 06-magD | 50-52 | 55.80 | 9.5 | 0.7 | 1.854 | 0.037 | 0.0062 | 0.0001 |
| 07-magA | 60-62 | 24.40 | 18.6 | 1.4 | 1.517 | 0.031 | 0.0148 | 0.0003 |
| 08-magA | 80-82 | 30.10 | 17.8 | 1.3 | 0.747 | 0.016 | 0.0287 | 0.0005 |
| 09-magA | 100-102 | 27.40 | 25.9 | 1.7 | 0.744 | 0.016 | 0.0419 | 0.0005 |
| 09-magC | 100-102 | 25.08 | 10.0 | 0.9 | 2.659 | 0.054 | 0.0045 | 0.0001 |
| 10-magA | 125-127 | 30.10 | 9.1 | 0.8 | 0.854 | 0.018 | 0.0128 | 0.0003 |
| 11-magA | 150-152 | 27.20 | 12.0 | 0.9 | 3.247 | 0.065 | 0.0045 | 0.0001 |
| 12-magA | 175-177 | 23.17 | 4.8 | 0.8 | 0.433 | 0.010 | 0.0134 | 0.0011 |
| 12-magB | 175-177 | 29.20 | 10.3 | 0.8 | 0.965 | 0.020 | 0.0128 | 0.0003 |
| 12-magF | 175-177 | 22.35 | 7.7 | 0.7 | 1.053 | 0.022 | 0.0087 | 0.0003 |




**Table 3: $^3$He and $^4$He concentrations of magnetite aliquots with inclusions, as confirmed by microCT.**

| Sample 17WW- | inclusions | depth [cm] | # of grains | mass [mg] | $^3$He conc. [Mat g$^{-1}$] | 1σ [Mat g$^{-1}$] | $^4$He conc. [nmol g$^{-1}$] | 1σ [nmol g$^{-1}$] | $^3$He/$^4$He [Ra] | 1σ [Ra] |
|---|---|---|---|---|---|---|---|---|---|---|
| 02-Incl1 | bright inclusions | 10-12 | 18 | 9.64 | 11.9 | 1.2 | 1.457 | 0.030 | 0.0098 | 0.0002 |
| 02-Incl2 | dark inclusions | 10-12 | 34 | 18.16 | 10.0 | 0.8 | 0.130 | 0.005 | 0.0930 | 0.0015 |
| 02-Incl3 | bright and dark inclusions | 10-12 | 19 | 13.88 | 12.2 | 1.1 | 1.234 | 0.025 | 0.0119 | 0.0002 |
| 02-Incl4 | many bright inclusions | 10-12 | 6 | 3.66 | 22.4 | 2.4 | 6.313 | 0.126 | 0.0043 | 0.0001 |
| 02-Incl5 | many dark inclusions | 10-12 | 6 | 6.88 | 12.8 | 1.4 | 0.085 | 0.005 | 0.1806 | 0.0045 |
| 02-Incl6 | many bright and dark inclusions | 10-12 | 12 | 7.22 | 13.7 | 1.4 | 0.779 | 0.016 | 0.0211 | 0.0005 |
| 03-Incl1 | many dark inclusions | 20-22 | 29 | 14.09 | 18.7 | 1.4 | 0.279 | 0.007 | 0.0805 | 0.0011 |
| 04-Incl1 | bright inclusions | 30-32 | 22 | 10.00 | 10.4 | 1.1 | 0.269 | 0.007 | 0.0467 | 0.0012 |
| 04-Incl2 | bright and dark inclusions | 30-32 | 18 | 7.82 | 16.3 | 1.6 | 1.529 | 0.031 | 0.0128 | 0.0003 |
| 04-Incl3 | many bright and dark inclusions | 30-32 | 18 | 12.56 | 30.6 | 2.2 | 0.892 | 0.018 | 0.0412 | 0.0005 |
| 09-Incl1 | many bright inclusions | 100-102 | 14 | 3.63 | 4.9 | 1.4 | 0.552 | 0.012 | 0.0108 | 0.0008 |
| 09-Incl2 | many bright inclusions, crushed | 100-102 | 15 | 7.62 | 6.4 | 0.9 | 1.396 | 0.028 | 0.0055 | 0.0002 |
| 09-Incl3 | many dark inclusions | 100-102 | 31 | 18.23 | 5.6 | 0.6 | 0.249 | 0.007 | 0.0269 | 0.0007 |
| 09-Incl4 | many dark inclusions, crushed | 100-102 | 37 | 15.94 | 7.8 | 0.8 | 0.356 | 0.008 | 0.0265 | 0.0006 |
| 11-Incl1 | many bright inclusions | 150-152 | 15 | 9.21 | 4.9 | 0.8 | 2.547 | 0.051 | 0.0023 | 0.0001 |
| 11-Incl2 | many bright inclusions, crushed | 150-152 | 23 | 10.02 | 6.8 | 0.8 | 0.920 | 0.019 | 0.0089 | 0.0005 |
| 11-Incl2 | many dark inclusions | 150-152 | 31 | 15.77 | 5.6 | 0.6 | 0.335 | 0.008 | 0.0202 | 0.0006 |
| 11-Incl3 | many dark inclusions, crushed | 150-152 | 41 | 11.92 | 5.9 | 0.7 | 0.308 | 0.008 | 0.0229 | 0.0006 |
| 11-Incl4 | many bright and dark inclusions | 150-152 | 30 | 16.10 | 8.2 | 0.8 | 0.978 | 0.020 | 0.0101 | 0.0002 |
| 11-Incl5 | many bright and dark inclusions, crushed | 150-152 | 30 | 16.35 | 6.2 | 0.6 | 1.329 | 0.027 | 0.0056 | 0.0001 |




**Table 4: $^3$He and $^4$He concentrations (conc.) in magnetite aliquots without inclusions, as confirmed by microCT. The $^3$He concentrations corrected (corr.) for nucleogenic and CTN-produced components are also given.**

| Sample 17WW- | depth [cm] | # of grains | mass [mg] | $^3$He conc. [Mat g⁻¹] | 1σ [Mat g⁻¹] | $^3$He corr. [Mat g⁻¹] | 1σ [Mat g⁻¹] | $^4$He conc. [nmol g⁻¹] | 1σ [nmol g⁻¹] | $^3$He/$^4$He [Ra] | 1σ [Ra] |
|---|---|---|---|---|---|---|---|---|---|---|---|
| 01N-A | 0-5 | 30 | 12.08 | 11.2 | 0.7 | 10.8 | 0.9 | 0.240 | 0.007 | 0.0564 | 0.0012 |
| 01N-B | 0-5 | 25 | 4.30 | 12.9 | 1.3 | 10.1 | 1.7 | 1.364 | 0.028 | 0.0113 | 0.0003 |
| 02N-A | 10-12 | 35 | 12.04 | 8.3 | 0.6 | 7.8 | 0.7 | 0.203 | 0.006 | 0.0490 | 0.0013 |
| 02N-B | 10-12 | 53 | 18.69 | 9.9 | 0.6 | 9.5 | 0.6 | 0.184 | 0.006 | 0.0650 | 0.0013 |
| 03N-A | 20-22 | 60 | 20.35 | 7.9 | 0.5 | 7.3 | 0.5 | 0.176 | 0.006 | 0.0542 | 0.0010 |
| 04N-A | 30-32 | 41 | 14.32 | 10.5 | 0.6 | 9.9 | 0.7 | 0.175 | 0.006 | 0.0722 | 0.0014 |
| 05N-A | 40-41 | 68 | 21.07 | 5.7 | 0.4 | 5.2 | 0.4 | 0.133 | 0.005 | 0.0512 | 0.0010 |
| 06N-A | 50-52 | 55 | 13.23 | 7.5 | 0.5 | 6.6 | 0.6 | 0.235 | 0.007 | 0.0385 | 0.0008 |
| 07N-A | 60-62 | 65 | 15.89 | 11.5 | 0.6 | 10.9 | 0.7 | 0.187 | 0.006 | 0.0743 | 0.0012 |
| 08N-A | 80-82 | 32 | 12.18 | 10.6 | 0.7 | 6.3 | 1.1 | 1.595 | 0.032 | 0.0080 | 0.0002 |
| 09N-A | 100-102 | 126 | 57.50 | 5.6 | 0.2 | 5.0 | 0.3 | 0.197 | 0.006 | 0.0343 | 0.0005 |
| 10N-A | 125-127 | 74 | 34.44 | 3.4 | 0.2 | 3.1 | 0.2 | 0.093 | 0.005 | 0.0434 | 0.0010 |
| 11N-A | 150-152 | 131 | 49.62 | 3.3 | 0.2 | 2.7 | 0.2 | 0.194 | 0.006 | 0.0203 | 0.0003 |
| 12N-A | 175-177 | 128 | 50.33 | 3.2 | 0.2 | 2.5 | 0.2 | 0.262 | 0.007 | 0.0148 | 0.0002 |
| 12N-B | 175-177 | 92 | 34.69 | 3.5 | 0.2 | 3.1 | 0.2 | 0.157 | 0.006 | 0.0265 | 0.0005 |