# Peer review of "Exposure dating of detrital magnetite using 3He enabled by microCT and calibration of the cosmogenic 3He production rate in magnetite"

_Geochronology, 2021_

## Referee Comment (RC1)

**Review of manuscript gchron-2021-10:**

**"Exposure dating of detrital magnetite using $^3$He enabled by microCT and calibration of the cosmogenic $^3$He production rate in magnetite"**

by F. Hofmann et al.

Comments by reviewer Samuel Niedermann

*General comment*

This manuscript proposes a new method to improve the accuracy of comsogenic $^3$He surface exposure dating using magnetite. The authors have used microCT to screen magnetite grains for the presence of inclusions. They argue that inclusion-free grains should be used for exposure studies, since grains with inclusions have frequently increased $^3$He concentrations due to several processes. Based on their He analyses of inclusion-free magnetite and a comparison to cosmogenic $^{10}$Be and $^{26}$Al concentrations in co-existing quartz, they propose a cosmogenic $^3$He production rate in magnetite of 116 at/g/a (scaled to sea level and high latitude).

The method proposed by the authors is clearly an important contribution to the wide field of surface exposure dating with cosmogenic nuclides, which has the potential to make some scientific questions accessible to research that have been difficult to treat so far, even though it is unlikely to become a standard method because of the various laborious analyses that are required. For the most part, the paper is written clearly, concisely and in good English. A few points need more explanation, as specified below, and an annoying point is that many references are missing in the reference list. The major flaw is the lack of discussion why some earlier production rate estimates were just about half the new rate (see below). Nevertheless, I recommend this manuscript for publication in *Geochronology* after minor revision has taken account of the specific and technical comments given hereafter.

*Specific comments:*

- At many places in the manuscript, the authors use expressions such as "cosmogenic production", "cosmogenic exposure", "cosmogenic studies" etc., or similar with radiogenic, nucleogenic. However, "cosmogenic" means "generated by cosmic rays" ("radiogenic" is "generated by radioactive decay" etc.), therefore one can only talk about cosmogenic nuclides, cosmogenic neutrons etc. but not about cosmogenic exposure or studies, and cosmogenic production is a pleonasm. I found such incorrect use of ...genic in the following lines: 16,17,38,45,71,85,87,90,92,309,380,422,425,429,432,440,454,462, 672.

- The term "radiogenic $^3$He" is a bit misleading. Usually, radiogenic means production by $\alpha$ or $\beta$ decay (or electron capture), while nuclides produces by fission are called fissiogenic. Even thogh fission is a kind of radioactive decay, I would suggest to use that distinction for $^3$He just like people do for Xe, because it makes clear that it is not about $^3$He production from $^3$H. If the authors still prefer radiogenic, they should explain what they mean at the first locations where this appears, i.e line 17 (Abstract) and 69 (main text).

- In lines 85-86, the authors mention a "high-energy muon component of spallogenic production", saying it is negligible. However, they don't mention slow (negative) muon capture at all. As the reference Nesterenok and Yakubovich 2016 is not shown in the reference list, I could not check whether these authors perhaps say slow muons are generally negligible for $^3$He. Even if so, this should be mentioned in the text.

- In line 112, "high-eU inclusions" are mentioned without an explanation what this means. The explanation follows much later (line 291), but even there no definition of effective uranium is given.

- "high-Ra helium" (line 118) is an incorrect expression. Ra ist the atmospheric $^3$He/$^4$He ratio, which is a fixed value, thus there is no low- or high-Ra helium. If anything like that, it should be high R/Ra, but preferably (and easier to understand for non-experts) I would suggest "helium with high $^3$He/$^4$He ratios".

- In line 132, an exposure age is given with an error shown with 3 significant digits. Even if this is taken from the reference, it is inappropriate to give more than two significant digits for an uncertainty, because uncertainties are not precise numbers but just represent probabilities. Also, values should always be given with the same precision as the corresponding uncertainties. Therefore, this value should be rounded to $54^{+19}_{-13}$ ka. Similarly, in line 391 should be 29.6±4.6.

- On a similar issue, a single 1 as the only significant digit (i.e. such as 1, 0.1, 0.01 etc.) is too little precision to show an error (Tables 2 and 3). E.g., 0.1 could have been rounded from anything between 0.05 and 0.14999, i.e. a factor of 3 difference in the actual precision of the measurement. In such cases it is necessary to give one more digit (for the corresponding value, too; see above).

- Perhaps some explanation of "isosurface renderings" (lines 224-225) would be appropriate (I don't know what this means).

- The term "$^3$He excess" in line 258 is misleading. Usually, in cosmogenic $^3$He literature, it is used for the excess of $^3$He over a typical He composition, such as mantle He, but here it obviously just means a higher $^3$He concentration in grains with inclusions. Such equivocal use of terms should be avoided.

- In line 316, the authors wrote "production of nucleogenic $^3$He from $^{10}$B is negligible". However, this process wasn't mentioned at all in section 2.1.

- In line 343, I don't unerstand why the combined RTN production rate is higher than the sum of the individual rates from U and Th.

- The method of correcting for different non-cosmogenic (or better: non-spallation-produced) components is generally clear, but I didn't understand (in lines 348-349) whether for each magnetite sample the nucleogenic $^3$He contribution was calculated based on its individual $^4$He closure age (but using mean U and Th concentrations) or whether a mean age was used for all samples. Anyway, these corrections have to be documented in a much better way. Rather than just showing uncorrected and corrected $^3$He concentrations in Table 4, each individual correction applied should be listed for each sample so the reader can retrace what the authors did. Also, there is no discussion at all about the estimated uncertainties of the corrections, though the higher uncertainties for corrected than uncorrected data show that some error estimate has been applied.

- Perhaps the major flaw of this manuscript, there is no discussion nor attempt of an explanation why the production rate of $^3$He in magnetite obtained here is almost a factor of two higher than previous model results and experimental determinations. Though the agreement with Kober et al. (2005) is excellent, it remains completely mysterious why Bryce and Farley (2002) obtained a much smaller rate (which agreed with Masarik and Reedy's model calculations). The presence of inclusions in Bryce and Farley's samples can obviously not explain the discrepance, as they would lead to an overestimate rather than an underestimate of the production rate, as shown in this manuscript. Therefore, without any argument why earlier estimates were so much lower, the production rate value reported here cannot really be considered reliable. Just ignoring the lower production rate estimates as done in the Conclusions (line 465) doesn't help.

*Technical comments:* (numbers refer to line numbers in the manuscript)

32 Calling the chemical procedures "dangerous" seems a bit strong. Of course HF (in particular) is a nasty substance, but using the appropriate precautions it can be handled routinely without being in permanent danger. So please, don't exaggerate!

68 Remove comma after "$^3$He data".

70 Change "cosmogenic magnetite $^3$He" to "magnetite cosmogenic $^3$He" (not the magnetite is cosmogenic, but the $^3$He).

236 Change "radiogenic" to "radioactive" (these elements are not products of radioactive decay, but they decay themselves).

240-241 Something wrong with a sentence; perhaps should be "Combined with … (Fig. 3), these data show ..."

313 Should be "production *of* $^3$He"

319 "to yield solely the cosmogenic component": Obviously what is meant is the component produced by cosmic ray spallation (+ muon interaction perhaps). The cosmogenic thermal neutron component is, however, cosmogenic too!

382 I assume this should be 1.7±0.6 Mat/g rather than at/g!

391-392 It should be stated that $^3$He is measured in magnetite and $^{10}$Be in quartz, and the ratio labeled $^3$He$_{mt}$/$^{10}$Be$_{qz}$ or similar.

460 "Knowledge of ... *is* important ..."

614 Remove dot after USA

655 inclusion*s*

695 "… of $^{10}$Be measured in …" (remove first "in")

Table 4: Please indicate whether the $^3$He/$^4$He ratio shown is the measured or corrected one. If measured, it would better be shown along with the other measured parameters, not after the corrected $^3$He.

Reference list: There are some inconsistencies in the referencing style (e.g. compare the first two entries). More importantly, the following references cited in the text cannot be found in the reference list:

Blackburn et al. 2007

Amidon et al. 2008

Nesterenok and Yakubovich 2016

Ziegler et al. 2010

Amidon and Farley 2009

Huerta 2017

Phillips et al. 2001

Gayer et al. 2004

In addition, Balbas and Farley 2020 should be before, not after Balco et al. 2008